# Genetic Testing for Familial Hypercholesterolemia in a Pediatric Group: A Romanian Showcase

**DOI:** 10.3390/diagnostics13121988

**Published:** 2023-06-07

**Authors:** Andreea Teodora Constantin, Ioana Streata, Mirela Silvia Covăcescu, Anca Lelia Riza, Ioana Roșca, Corina Delia, Lucia Maria Tudor, Ștefania Dorobanțu, Adina Dragoș, Diana Ristea, Mihai Ioana, Ioan Gherghina

**Affiliations:** 1Pediatrics Department, National Institute for Mother and Child Health “Alessandrescu-Rusescu”, 020395 Bucharest, Romaniasilvia.covacescu@umfcd.ro (M.S.C.); corina_delia@yahoo.com (C.D.); dr.luciatudor@gmail.com (L.M.T.); 2Pediatrics Department, Faculty of Medicine, University of Medicine and Pharmacy “Carol Davila”, 020021 Bucharest, Romania; prof_ighe@yahoo.com; 3Genetics Department, University of Medicine and Pharmacy, 200349 Craiova, Romaniastefania.crgm@gmail.com (Ș.D.); adina.crgm@gmail.com (A.D.); mihai.ioana@umfcv.ro (M.I.); 4Regional Center for Medical Genetics Dolj, 200642 Craiova, Romania; diana.crgm@gmail.com; 5Faculty of Midwifery and Nursery, University of Medicine and Pharmacy “Carol Davila”, 020021 Bucharest, Romania; ioana.rosca@umfcd.ro; 6Neonatology Department, Clinical Hospital of Obstetrics and Gynecology “Prof. Dr. P.Sârbu”, 060251 Bucharest, Romania; 7Faculty of Biology, University of Bucharest, 030018 Bucharest, Romania

**Keywords:** hypercholesterolemia, genetic, pediatrics

## Abstract

Familial hypercholesterolemia (FH) is a genetic disease marked by high levels of LDL-cholesterol. This condition has long-term clinical implications, such as cardiovascular events, that are evident during adult life. Here, we report on a single-center cross-sectional showcase study of genetic testing for FH in a Romanian pediatric group. Genetic testing for FH was performed on 20 Romanian pediatric patients, 10 boys and 10 girls, admitted with LDL-cholesterol levels over 130 mg/mL to the National Institute for Mother and Child Health “Alesssandrescu-Rusescu” in 2020. Genetic testing was performed using the Illumina TruSight Cardio panel. We identified pathogenic/likely pathogenic variants that could explain the phenotype in 5/20 cases. The involved genes were *LDLR* and *APOB*. Clinical signs that suggest the diagnosis of FH are scarce for the pediatric patient, although it can be diagnosed early during childhood by lipid panel screening. Prevention could prove lifesaving for some of these patients.

## 1. Introduction

Familial hypercholesterolemia (FH) is a genetic disease characterized by high levels of LDL-cholesterol and a long-term risk of cardiovascular events [1]. The estimated global prevalence of FH is 1:250 individuals [2,3,4,5], with an incidence of homozygous FH varying between 1:160,000 and 1:300,000 and 1:1,000,000 [6]. It is thought to be highly underdiagnosed, especially in countries without active screening strategies in place [7]. Currently, Romania does not implement active screening strategies for FH.

FH has duly been investigated in the adult population; however, the pediatric patient cannot subscribe to similar diagnoses and management strategies. In children, the condition often goes unnoticed. The suggestive clinical signs are few and not always present: xanthomas, xanthelasmas and corneal arcus to name several [8], with the most common sites for tendon xanthomas being the extensor tendons of the dorsum of the hand and the Achilles tendon [9,10].

However, most studies recommend screening for FH before adolescence. All the more, since screening for the condition is easily available for clinicians, it requires taking an accurate familial history focused on the evidence of cardiovascular events at young ages [11] and an assessment of the cholesterol and LDL-cholesterol levels [12]. There is no international consensus regarding the level for LDL-cholesterol suggestive for FH; different studies and protocols recommend values over 140 mg/dL to over 190 mg/dL, depending on the country and author [7,11,13,14].

Genetic testing confirms the diagnosis but is not mandatory according to current recommendations. Some studies arguably advise against using genetic testing for the screening of FH [15]. Nonetheless, most FH cases have an identifiable genetic cause. Frequently, three main autosomal dominant genes are involved: the subtilisin-kexin type 9 protein convertase gene (*PCSK9*), the gene for LDL-cholesterol ligand apolipoprotein B 100 (*APOB*), and the gene for LDL-receptor (*LDLR*) [16,17]. Autosomal recessive inheritance has been described and is caused by variants in the LDL-receptor adapter protein (*LDLRAP*) gene [17]. 

Early diagnosis is a must; it translates into early interventions and improved outcomes. The first steps to take for the management of FH are lifestyle interventions to lower cardiovascular risk, which may include dietary recommendations, quitting smoking, regular physical activity and maintaining a healthy body weight [18,19]. Pharmacological options to treat FH include statins as the first line of therapy or other lipid-modifying drugs such as fibrates, resins, cholesterol reabsorption inhibitors, nicotinic acid derivates, small-interfering-RNA-based therapeutic and monoclonal antibodies [20]. 

The aim of this study, among the first in our country, is to identify genetic variations responsible for the high cholesterol levels seen in a group of pediatric Romanian patients with a clinical diagnosis FH and to use this study as a showcase for conducting molecular testing in extensive cohorts of pediatric patients. 

## 2. Materials and Methods

The study was approved by the ethics committee of the National Institute for Mother and Child Health “Alesssandrescu-Rusescu” (approval no. 12747/16.07.2020) and was conducted respecting the Declaration of Helsinki for human rights. For each child enrolled in the genetic testing group, informed consent was obtained from both parents of the child.

We conducted a cross-sectional study at the National Institute for Mother and Child Health “Alesssandrescu-Rusescu” in Bucharest, Romania. Thus, 20 patients of the institute between 2011 and 2019 (and revisiting the clinic in 2020) were included to undergo genetic testing for FH. In our study group, the patients were equally distributed between sexes and the mean age was 9.70 (±4.09) years old. 

The inclusion criteria were: Romanian ethnicity, age under 18 years old, LDL-cholesterol level over 130 mg/dL, no other chronic disease that could explain dyslipidemia and parental agreement for participation in the study. We did not take into consideration whether the included patients received any kind of cholesterol-lowering drugs any time before or after the date of the lipid panel evaluation. 

Clinical evaluation included anthropometric measurements (weight and height) and blood pressure. Body mass index (BMI) was calculated according to the Center for Disease Control (CDC) international references. We classified each patient according to the BMI percentile as either underweight, healthy weight, overweight or obesity class I, II or III. CDC growth charts were used. Blood test panels included lipid evaluation. 

Genetic testing was performed by the Regional Center for Medical Genetics in Dolj, Craiova, Romania. DNA was isolated from the collected EDTA peripheral blood. Library preparation occurred as per manufacturer instructions. The next generation sequencing (NGS) panel of choice was TruSight Cardio (Illumina, San Diego, CA, USA). The kit covers exons and adjacent intronic regions for 174 genes known to be associated with hereditary heart and blood vessel diseases, including familial hypercholesterolemia. Sequencing was performed on NextSeq550Dx, aiming for a median of 100× coverage. Paired-end 2 × 150 bp reads were mapped to the human genome reference sequence (GRCh37, iGenomes resource bundle) The nf-core/sarek 2.7.1 pipeline was used [21]. The identified variants were annotated using an ENSEMBL variant effect predictor (VEP) [22,23]; online databases such as OMIM, ClinVar and Varsome were also consulted [24,25,26]. Annotated and inheritance information, where available, were used for ACMG compliant variant classification [27]. The coverage of several relevant genes was manually investigated. 

Descriptive and statistical analysis was performed using Epi Info™ For Windows, version 7.2. Parametrical or non-parametrical tests used considered the significance level threshold at 0.05.

## 3. Results

To give context to our showcase genetic study, we conducted a retrospective survey in our institution. Between 2011 and 2019, there were 1728 unique patients who had their lipid panel evaluation in our clinic, either as an inpatient or an outpatient. A total of 14.98% (259 patients) had LDL-cholesterol levels over 130 mg/dL (the threshold level for considering FH according to the Romanian Society of Pediatrics Protocols) [28]. We estimated that approximately 40–45% of these 259 patients met the criteria for FH after excluding other disorders that could interfere with cholesterol metabolism and result in high LDL-cholesterol levels. 

Genetic testing was offered as described above to 20 patients of this group, who revisited the clinic in 2020. We looked at the study group from the perspective of the genetic testing results and we considered positive testing as the presence of a pathogenic, likely pathogenic or variants of unknown significance (VUS).

### 3.1. Genetic Testing Results

Genetic testing was performed on 20 pediatric patients included in this study. The relevant variants identified are summarized in Table 1.

In 8/20 cases (40%), we could identify a variant that led to an ethio-pathogenical diagnosis of FH: four patients had heterozygous pathogenic or likely pathogenic variants on the *LDLR* gene (type 1 familial hypercholesterolemia), two patients had variants of unknown significance on the *PCSK9* gene (type 3 familial hypercholesterolemia) and two patients had heterozygous mutations on the *APOB* gene (type 2 familial hypercholesterolemia).

### 3.2. Clinical Data

There was no difference regarding the distribution of sex or age between the positive and negative genetic testing subgroups. Clinical data are summarized in Table 2.

Clinical examination did not reveal any clinical sign suggesting FH for any of the patients, even for those in the genetic-positive group. 

Only 16.67% (3/20) of the patients had a healthy weight. A total of 38.89% (7/20) of the patients included in the study were underweight, while 22.22% (4/20) had class I obesity. BMI values were comparable between the genetic-positive group (17.57 ± 6.26) and the genetic-negative group (20.44 ± 8.10). In the genetic-positive group, the patients were underweight, overweight or had class I obesity. Interestingly, one case of a girl with class III obesity according to her age and sex had genetic testing that was negative for FH.

Blood pressure values were classified according to the patient’s age, sex and height according to the American Academy of Pediatrics clinical practice guidelines: normal blood pressure, elevated blood pressure (EBP), stage I (SIHBP) or II high blood pressure (SIIHBP). There were seven patients with normal blood pressure (five of whom were in the genetic-positive group) and four patients with EBP (three of whom were in the genetic-negative group). The patients with SIHBP and SIIHBP were mostly from the genetic-negative group. The classification of these patients’ blood pressure may change over time and follow-up separate measurements at each visit to the doctor would be required [29]. Mean systolic blood pressure (SBP) and diastolic blood pressure (DBP) in the genetic-positive group (108.62 mmHg ± 19.21; 62.50 mmHg ± 10.35) was relatively lower than the genetic-negative group (120.80 mmHg ± 12.50; 70.10 mmHg ± 6.43). 

Lipid profiles are summarized in Table 3. Total cholesterol in the study group had a mean value of 224.35 mg/dL (±30.75), while mean LDL-cholesterol was 154.25 mg/dL (±26.79). Total cholesterol level was slightly higher in the genetic-positive group (241.12 mg/dL ± 36.29) compared with the genetic-negative group (213.16 mg/dL ± 21.31). The same was observed for the LDL-cholesterol level: 165.12 mg/dL (±34.01) in the genetic-positive group and 147.00 mg/dL (±18.98) in the genetic-negative group. ApoA mean values were 149.05 mg/dL (±25.53 mg/dL). The ApoB mean values were 117.77 mg/dL (±17.54 mg/dL).

## 4. Discussion

FH is a frequent genetic disorder characterized by markedly elevated LDL-cholesterol levels from birth onward, resulting in a significantly increased risk of atherosclerotic cardiovascular disease (ASCVD). Despite all the progress, FH remains underdiagnosed and undertreated, leading to premature morbidity and mortality due to atherosclerotic cardiovascular disease [30]. 

The World Health Organization declared FH a public health priority in 1998. Affecting every race and ethnicity, FH is one of the most common genetic disorders in the world [31,32]. The prevalence of FH may vary widely between countries because of founder effects, use of different diagnostic criteria and differences in strategies for the identification of high-risk subgroups and screening for the disease [5,32]. However, no previous study has summarized the differences in the prevalence of FH among ethnic groups [31,33]. A Danish study conducted on the general population estimated the prevalence of FH as high as 1:137 [34]. It is likely that, in a general population setting, FH is underdiagnosed. This further emphasizes the importance of efficient FH screening to identify individuals at risk [32]. FH diagnosis is usually made late in life and only 2% of patients are diagnosed before the age of 18 [35]. Scientific new advances in genetic testing have made NGS more available worldwide and the costs of genetic testing have reduced [36,37,38].

Screening for FH can improve the low diagnostic rates from different countries. There are different screening methods that can be used, such as selective screening, cascade screening and universal screening. The most commonly used method in Europe is testing the children of parents diagnosed with FH [39,40]. Universal screening was used in Slovenia in 1995 [41] and cascade screening was used successfully in the Netherlands in 1994 [42]. In the United Kingdom, in pediatric patients aged been 1 and 2 years of age, a combination of universal cholesterol screening followed by genetic testing and reverse cascade testing was used [43]. Such an approach has multiple advantages, such as the possibility to implement preventive lifestyle choices as early as possible.

There are multiple clinical tools that can be used to diagnose FH, e.g., the Simone Broom Register Diagnostic Criteria [44] and the Dutch Lipid Clinic Network Criteria [45] are recommendations from the 2015 American Heart Association statement on FH [46]. These tools take genetic testing into consideration. According to the *Journal of the American College of Cardiology* expert panel recommendations, genetic testing should be offered to children with persistent LDL-cholesterol levels ≥160 mg/dL, without an apparent cause of hypercholesterolemia, who have at least one first-degree relative who also has hypercholesterolemia or premature cardiovascular disease (or if the family history is not available) and to children with persistent LDL-cholesterol levels ≥190 mg/dL, regardless of their family history [38]. Even if a patient is diagnosed by clinical criteria, in 20% of cases there is no identifiable causative mutation in any of the previously known FH-associated genes [47].

The need for genetic testing in diagnosing FH is emphasized by the lack of clinical signs in children; xanthomas, xanthelasmas and corneal arcus are usually not present at this age [48]. Although some studies suggest that clinical criteria are sufficient to initiate treatment for FH [35,49,50], in certain countries, such as Romania, where there is scarce genetic information about this disorder, genetic testing would be helpful in determining population characteristics and genetic mutations specific to our population.

We may ask ourselves about the cost-effectiveness of screening for FH. In the United Kingdom, screening for FH was proposed to parents when they went to the doctor with their child for routine immunizations at the age of 1–2 years. Eighty-four percent of the parents agreed with the additional bloodwork [12,51]. Economics studies that estimated the cost-effectiveness of implementing a screening program for FH proved that, for every 1000th person tested, 46 heart attacks, 50 cases of angina, 8 strokes and 16 deaths could be avoided over a 20-year period. Additionally, the cost per quality-adjusted life year (QALY) index is 57% higher in patients diagnosed after the age of 55 years compared with the ones diagnosed between 20 and 54 years of age [52]. A study that took place in Spain [53] estimates that the incremental cost effectiveness is EUR 26,792 per avoided coronary event and EUR 111,567 per avoided death.

In countries where there is no screening program and no registry for FH, the tendency is to diagnose and treat FH later in life, after the cardiovascular event has already happened. A more preventive approach by early diagnosis (based on clinical and paraclinical, as well as genetic, criteria since childhood), addressing modifiable risk factors and genetic cascade screening can significantly decrease the morbidity and mortality of these patients [54].

Early genetic diagnosis of FH should be recommended before the initiation of lipid-lowering therapies. According to a large study that followed therapy in children for 20 years [55], early treatment greatly reduced or delayed the risk of cardiovascular complications in adulthood. To initiate appropriate treatment, it is important to know whether the patient has a heterozygous, compound heterozygous or homozygous genotype. In addition, by cascade testing, the affected yet asymptomatic family members can be identified, leading to correct and timely treatment [56]. 

When talking about the need for genetic testing for FH, it has been shown that genetic confirmation supports the decision to start medication and has a positive effect on adherence to treatment and in lowering LDL-cholesterol levels [38]. From a patient’s perspective, it helps knowing that their high cholesterol levels are not related to behavior, that they know the cause and that effective treatment is available [39].

More than 2000 variants have been identified in patients with clinical FH [57]. Although FH genetic diagnosis is available nowadays all over the world, only Spain, the Netherlands and Uruguay have governmental approval for the identification of FH patients [58,59]. Our study is among the first studies carried out in Romania to contribute to FH early genetic diagnosis in patients with this condition. In our study, an NGS-based technique was used to identify variants in pediatric patients with a clinical diagnosis of FH. To our knowledge, no study describing the variants related to FH in the pediatric Romanian population has been published. 

In our study group, the detection rate was 40%: we identified variants in eight cases; five patients had pathogenic/likely pathogenic variants and three had VUS. Strikingly, the pathogenic/likely pathogenic variants were mainly identified in the LDLR gene. Studies that identified mutations also present in our patients are summarized in Table 4.

The *LDLR* c.1618G>A, p.(Ala540Thr) missense variant, detected in one index patient, is also a commonly occurring variant among Romanian patients and has been described in several European, Latin American and even Chinese populations [56,60,61,62]. It is located in the epidermal growth factor (EGF)-precursor homology domain. Its pathogenicity is demonstrated by the fact that homozygous patients present with severe clinical symptoms at a young age, e.g., premature coronary heart disease, tendon xanthomas or high total cholesterol level [56]. The phenotype of our patient was similar to the literature-reported clinical presentation in terms of high cholesterol level, but he did not show any other clinical signs. 

In one case, we identified a heterozygous missense *LDLR* c.1775G>A, p.(Gly592Glu). This variant was associated with a strong founder effect in a Greek region, where it was present in almost one-third of the patients examined and detected in a homozygous form in several cases [63]. According to the literature data, the cholesterol-elevating effect does not associate with its zygosity [56]. In addition, nonsense variations have been previously described as FH-causing pathogenic variants. These result in a truncated form of the LDLR protein, which is nonfunctional. The uptake of LDL particles is reduced in the liver and other tissues.

Due to next-generation sequence techniques being more available, different types of mutations in the *APOB* gene are emerging as relevant for FH. In our study, two patients had *APOB* mutation (one likely pathogenic and one variant of unknown significance). In another Romanian study, there was only one *APOB* mutation identified (c.10740C>T) [62]. This mutation was not present in our study. In the *APOB* gene, the most common variant in East European populations is the c.10580G>A p.(Arg3527Gln) missense change [64,65,66]. 

*PCSK9* mutations are usually less frequently identified in patients with FH [67]. In our study, there were two patients with the same *PCSK9* mutation. NM_174936.4:c.836C>T, currently assessed as a variant of unknown significance, was not identified in the Romanian study published in 2021 [62].

Another recent study, describing the variants related to FH in the Romanian adult population, has been published [62] and identifies three out of the four mutations on the *LDLR* gene that we report (c.1618G>A, c.502G>A and c.81C>G). Variants c.1618G>A and c.81C>G are also mentioned in several Chinese study groups [60,61]. There is a large overlap between Romania and its neighboring countries as far as the mutation spectrum goes. The most common variants in the *LDLR* gene in the Hungarian population can also be found in our cohort [56]. Our study revealed no novel variant in the *LDLR* gene; all the variants identified in the Romanian population have already been described. The proportion of variant types and their localization within the gene are also highly similar in population studies. The concern for other ethnicities lies in the need for the medical healthcare system to address all patients.

Treatment for FH aims to lower the LDL-cholesterol levels of patients and delay or even prevent cardiovascular disease in patients. The risk of developing cardiovascular disease is increased twenty-two-fold when a mutation implicated in FH is detected [77]. The first step of treatment is recommending a diet with low-saturated fat and supporting physical activity [78]. If the LDL-cholesterol level desired is not achieved, medical treatment with statins can be started. Adding ezetimibe to the statin treatment can lower the LDL-cholesterol levels by a further 24% [2,46]. Treatment with statins is approved for pediatric patients starting from 6 to 10 years of age in Europe, while ezetimibe can be added if the patient is older than 10 [46,79,80,81]. *PCSK9*-based therapies have been a real game changer in the treatment of FH; they can be used if the LDL-cholesterol level remains high or above target after maximal treatment with statins and ezetimibe [79,82]. Although initially used in adults and patients with homozygous FH, the use of *PCSK9* inhibitors has increased in children and patients with heterozygous FH [83]. For adults with FH, there are also other medical options such as lomitapide (inhibitor of the microsomal triglyceride transfer protein) and mipomersen (antisense oligonucleotide inhibitor targeted to apolipoprotein B-100 mRNA) [84,85,86,87]. Lipoprotein apheresis is the last line of therapy; it is not available worldwide, it is expensive and the eligibility criteria are different from country to country. The recent increase in the use of the *PCSK9* inhibitor has lowered the need for lipoprotein apheresis use [88,89,90]. 

For our study, certain limitations were inherent as the number of patients included in the study was small. This was due especially to the restrictive cost and availability of state-funded genetic testing for FH. 

Additionally, this study was conducted as a single-center study as a means to showcase the potential of molecular testing. The lack of genetic studies on FH in our country does not allow us to correlate our findings with the local geographic spread of mutations; further studies are warranted, including studies in other ethnicities. 

Other limitations of our study that might also be considered: (i) subjective assessment of some clinical features cannot be excluded; (ii) the lack of extended family targeted testing of identified variants by Sanger sequencing; (iii) the inheritance status is missing for the interpretation of VUS variants; (iv) low-level mosaicism cannot be detected.

## 5. Conclusions

In this study, 8 out of 20 patients had positive-genetic testing for FH. We identified five patients with pathogenic/likely pathogenic variants and three had variants of uncertain significance. Most frequent mutations identified concerned *LDLR,* but *APOB and PCSK9* variants were identified. Several variants had been previously reported (*LDLR* c.1618G>A, *LDLR* c.1775G>A).

Evaluation of the lipid profile, which is the first step towards the clinical diagnosis of FH, is universally available and inexpensive. Genetic testing remains the gold standard for FH diagnosis [91]. Studies have demonstrated the cost-effectiveness of screening for FH, early detection and early intervention for decreasing cardiovascular disease incidence among affected individuals [91]. 

Treatment for FH starts with lifestyle changes. If goal LDL-cholesterol levels are not achieved with lifestyle changes alone, statins can be used. Additional medication options include ezetimibe, *PCSK9* therapies, lomitapide and ultimately lipoprotein apheresis. Lipoprotein apheresis is not available worldwide and the need for its use has decreased significantly since the use of *PCSK9* inhibitors has increased.

Taking into consideration the high morbidity and mortality of patients with FH when exposed to risk factors, early diagnosis in childhood (when there are scarce to no clinical signs) could prevent cardiovascular events and increase the quality of life.

Further studies to include larger number of pediatric patients are recommended, targeting the identification of the genetic spectrum of FH in Romania and other populations. 

## Figures and Tables

**Table 1 diagnostics-13-01988-t001:** Variants identified in our Romanian pediatric study group.

Case Number	FH	Affected Gene	Variant		ACMG Variant Classification
#4	Type 2 OMIM #144010 AD *	*APOB*	NM_000384.3:c.10580G>A rs5742904 (p.Arg3527Gln)	Heterozygous	Likely pathogenic
#3	Type 2 OMIM #144010 AD	*APOB*	NM_000384.3:c.12443_12444delinsAA rs1558559244 (p.Ala4148Glu)	Heterozygous	VUS
#2	Type 1 OMIM #143890 AR */AD	*LDLR*	NM_000527.5:c.1618G>A rs769370816 (p.Ala540Thr)	Heterozygous	Pathogenic
#6	Type 1 OMIM #143890 AD/AR	*LDLR*	NM_000527.5:c.1775G>A rs137929307 (p.Gly592Glu)	Heterozygous	Pathogenic
#7	Type 1 OMIM #143890 AD/AR	*LDLR*	NM_000527.5:c.502G>A rs200727689 (p.Asp168Asn)	Heterozygous	Pathogenic
#8	Type 1 OMIM #143890 AD/AR	*LDLR*	NM_000527.5:c.81C>G rs2228671 (p.Cys27Trp)	Heterozygous	Likely pathogenic
#1	Type 3 OMIM #603776 AD	*PCSK9*	NM_174936.4:c.836C>T rs1049662014 (p.Pro279Leu)	Heterozygous	VUS
#5	Type 3 OMIM #603776 AD	*PCSK9*	NM_174936.4:c.836C>T rs1049662014 (p.Pro279Leu)	Heterozygous	VUS

* AD—autosomal dominant, AR—autosomal recessive.

**Table 2 diagnostics-13-01988-t002:** Clinical data for the study group, the genetic-positive group and the genetic-negative group. No statistically significant differences were identified.

	Study Group (*n* =20)	Negative Genetic Testing (*n* = 12)	Positive Genetic Testing (*n* = 8)	*p* Values
Gender	10 girls 10 boys	7 girls 5 boys	3 girls 5 boys	
Positive family history	10 cases	7 cases	3 cases	0.32
Mean age	9.70 years (±4.09)	10.16 years (±4.58)	9.00 years (±3.38)	0.43
BMI	19.49 (±7.48)	20.44 (±8.10)	17.57 (±6.26)	1.00
Mean systolic blood pressure	115.38 mmHg (±16.53)	120.80 mmHg (±12.50)	108.62 mmHg (±19.21)	0.22
Mean diastolic blood pressure	66.72 mmHg (±9.00)	70.10 mmHg (±6.43)	62.50 mmHg (±10.35)	0.09

**Table 3 diagnostics-13-01988-t003:** Lipid profile results for the study group, the genetic-positive group and the genetic-negative group. No statistically significant differences were identified.

	Study Group (*n* = 20)	Negative Genetic Testing (*n* = 12)	Positive Genetic Testing (*n* = 8)	*p* Values
Total cholesterol	224.35 mg/dL (±30.75)	213.16 mg/dL (±21.31)	241.12 mg/dL (±36.29)	0.07
LDL-cholesterol	154.25 mg/dL (±26.79)	147.00 mg/dL (±18.98)	165.12 mg/dL (±34.01)	0.24
HDL-cholesterol	59.55 mg/dL (±17.69)	54.41 mg/dL (±12.51)	67.25 mg/dL (±22.14)	0.16
ApoA	149.05 mg/dL (±25.53)	139.30 mg/dL (±21.72)	161.25 mg/dL (±32.64)	0.14
ApoB	117.77 mg/dL (±17.54)	114.40 mg/dL (±10.80)	122.00 mg/dL (±23.68)	0.39

**Table 4 diagnostics-13-01988-t004:** Different studies that reported mutations identified in our patients.

Affected Gene	Variant	Country	Number of Patients Included in the Study	Technique Molecular Analysis
*LDLR*	NM_000527.5:c.1775G>A	Greece [63]	73	Direct DNA sequencing of LDL gene (previously identified LDL mutation)
*LDLR*	NM_000527.5:c.502G>A	Romania [62]	61	Multiplex ligation-dependent probe amplification (MLPA) Sanger sequencing
		Czech Republic-1 [64]	3914	MLPA
*LDLR*	NM_000527.5:c.81C>G	Romania [62]	61	MLPA Sanger sequencing
		China-1 [61]	208	MLPA
*LDLR*	NM_000527.5:c.1618G>A	Romania [62]	61	MLPA Sanger sequencing
		Hungary [56]	44	MLPA
		Poland [65]	161	MLPA
		Greece [68]	183	Denaturing gradient gel electrophoresis (DGGE) analysis
		Spain [69]	476	Single-strand conformation polymorphism (SSCP) analysis
		Germany [70]	162	MLPA Direct sequencing on LDL gene
		United Kingdom [71]	280	MLPA Sequencing of amplified fragments of genomic mRNA or DNA
		Brazil [72]	248	MLPA
		Australia [73]	30	MLPA Sanger sequencing Ion torrent personal genome machine (PGM) sequencing
		Taiwan [74]	125	Sanger sequencing Microarray resequencing
		Japan [75]	205	MLPA SSCP assay
		China-2 [60]	377	MLPA
		China-1 [61]	208	MLPA
*APOB*	NM_000384.3:c.10580G>A	Czech Republic-2 [76]	2239	MLPA
		Poland [65]	161	MLPA

## Data Availability

Not applicable.

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
