# Peer review of "Genetic Testing for Familial Hypercholesterolemia in a Pediatric Group: A Romanian Showcase"

_diagnostics, 2023, doi:10.3390/diagnostics13121988_

Round 1

Reviewer 1 Report

This paper is a small pilot study that has investigated the genetics of Familial Hypercholesterolemia in Romania.  They authors have examined 174 genes for potential mutations and successfully identified 8 variants using next generation sequencing. The authors have acknowledged the limitations (which I agree with).

The research team have mined through a huge volume of nucleotide data to search for variants and generated some interesting data that would benefit with publication. The identification and sharing of VUS and Class 4/5 mutations (no matter how small the sample size is) is really important, particularly as more genetic testing is taking place and clinicians are using pharmacogenomic information for treatment.

The authors have provided sufficient background, methodology, results and discussion. The methods were very concise and reproducible to a certain degree. The paper was also supplemented with appropriate citations.

I enjoyed reading your discussion section. The scientific argument in this sections was strong, informative and well written. 

Minor corrections:

Line 16 - correct spelling mistake. "Gentics"

Formatting - use italics for gene symbols

Line 32 - should it be: " LDLR or APOB"

Line 42 - unclear if Romania has active screening strategies for FH.

Line 72 - Is there any published data regarding genetic variation in FH genes in pediatric Romanian patients? Is this the first study of its kind for  Romania? Yes, you mention this later in the discussion. I would recommend that this is stated earlier in the introduction . 

Line 82 - insert a comma "Bucharest, Romania"

Clarification - Are the 20 patients recruited of Romanian ethnicity or live in Romania and are from other ethic groups (later your raise that there are no studies regarding ethnic groups). Unclear.

Line 101 - Include DNA source and concentration? 

2. Materials and Methods - in the results you mention p-values but no mention of what statistical method and programme was used. Also, what p value you consider is significant. Please include this in Section 2.

Line 130 - Table 1. Unclear the order. I would have grouped with gene name or FH type rather than by case n#.

Table 2. Remove comma's and add full stops. i.e, assuming mean years is 9.70 years 

Line 242: Replace capital letter with low case  "the Netherlands"

Line 308: Insert a space between full stop and letter.  ".For"

Reference #11. Include doi: link

Author Response

Dear reviewer,

We are grateful for the time and effort that you and the other reviewers dedicated to providing feedback on our manuscript and are thankful for the insightful comments and valuable improvements to our paper.

We have made all efforts to address all the concerns you raised. We have incorporated most of the suggestions made. The changes can be found in the track-changes new manuscript. Please see below for a point-by-point response to your comments and concerns, marked in blue.

Best wishes,

The authors.

The authors have provided sufficient background, methodology, results and discussion. The methods were very concise and reproducible to a certain degree. The paper was also supplemented with appropriate citations.

I enjoyed reading your discussion section. The scientific argument in these sections was strong, informative and well written. 

Minor corrections:

Line 16 - correct spelling mistake. "Gentics"

Thank you, it is now corrected.

Formatting - use italics for gene symbols

Thank you, it was an oversight we have now corrected.

Line 32 - should it be: " LDLR or APOB"

Thank you.

Line 42 - unclear if Romania has active screening strategies for FH.

Thank you, we added a clear statement following this paragraph – Romania does not have active screening strategies.

Line 72 - Is there any published data regarding genetic variation in FH genes in pediatric Romanian patients? Is this the first study of its kind for  Romania? Yes, you mention this later in the discussion. I would recommend that this is stated earlier in the introduction . 

Thank you, we have also mentioned our study is“among the first in Romania” in the introduction.

Line 82 - insert a comma "Bucharest, Romania"

Thank you, resolved.

Clarification - Are the 20 patients recruited of Romanian ethnicity or live in Romania and are from other ethic groups (later your raise that there are no studies regarding ethnic groups). Unclear.

Thank you for your observation. The 20 patients included in the study were ethnically Romanian; we have added the information under inclusion criteria in the methods section. Hungarian and Rroma are 2 major ethnicities in Romania which also should be addressed in genetic studies, we feel.

Line 101 - Include DNA source and concentration? 

Thank you. We have now added several clarifying sentences.

  1. Materials and Methods - in the results you mention p-values but no mention of what statistical method and programme was used. Also, what p value you consider is significant. Please include this in Section 2.

Thank you, this has now been addressed in an additional paragraph in the closing of the methods section.

Line 130 - Table 1. Unclear the order. I would have grouped with gene name or FH type rather than by case n#.

Thank you, the table has now been rearranged to reflect per gene variants.

Table 2. Remove comma's and add full stops. i.e, assuming mean years is 9.70 years 

Thank you, I believe these were journal recommendations as far as decimal separator goes.

Line 242: Replace capital letter with low case  "the Netherlands"

Thank you, this has now been corrected.

Line 308: Insert a space between full stop and letter.  ".For"

Thank you, this has now been corrected.

Reference #11. Include doi: link

Thank you, this has now been added.

Reviewer 2 Report

The study is interesting because the authors analyzed rare gene variants associated with lipid metabolism in patients with familial hypercholesterolemia in a pediatric group in Romania.

 It is necessary to include in table 1 a column with the designation of variants at the protein level and a column with the dbSNP ID.

Did statistically significant differences were identified family history results for the study group, the genetic-positive group and the genetic-negative group?

Have you analyzed the isolation of pathogenic variants with a pathological phenotype in pedigrees?

Have you identified any Pathogenic or Likely-Pathogenic variants from other genes (ABCC5, ABCC8 et al.)?

Author Response

Dear reviewer,

We are grateful for the time and effort that you and the other reviewers dedicated to providing feedback on our manuscript and are thankful for the insightful comments and valuable improvements to our paper.

We have made all efforts to address all the concerns you raised. We have incorporated most of the suggestions made. The changes can be found in the track-changes new manuscript. Please see below for a point-by-point response to your comments and concerns, marked in blue.

Best wishes,

The authors.

The study is interesting because the authors analyzed rare gene variants associated with lipid metabolism in patients with familial hypercholesterolemia in a pediatric group in Romania.

It is necessary to include in table 1 a column with the designation of variants at the protein level and a column with the dbSNP ID.

Your suggestion has been taken on board, thank you. Table 1 now includes the recommended denominations. We do feel the rs ID is to be avoided due to possible confusion.

Did statistically significant differences were identified family history results for the study group, the genetic-positive group and the genetic-negative group?

Thank you for your suggestion. We have refrained from making strong statements about statistical significance given the small numbers in our cohort, but we have now presented family history in table 2.

Have you analyzed the isolation of pathogenic variants with a pathological phenotype in pedigrees?

Thank you for raising this question. Nonetheless, the majority of the identified variants were reported to have autosomal dominant inheritance. Detecting one individual with an AD pathogenic variant may lead to a family intervention. We agree it would have been an interesting information to have, unfortunately, not all parents consented to the genetic testing at this time.

Have you identified any Pathogenic or Likely-Pathogenic variants from other genes (ABCC5, ABCC8 et al.)?

Thank you for your question. ABCC9, ABCG5, ABCG8  are part of the Cardio panel and were evaluated https://emea.illumina.com/content/dam/illumina-marketing/documents/clinical/rgh/gene-list.xlsx . No variants have been found, other than those reported.

Round 2

Reviewer 2 Report

The study is interesting because the authors analyzed rare gene variants associated with lipid metabolism in patients with familial hypercholesterolemia in a pediatric group in Romania. The publication can be accepted in present form.